# Diagnosis of Central Sensitization and Its Effects on Postoperative Outcomes following Total Knee Arthroplasty: A Systematic Review and Meta-Analysis

**DOI:** 10.3390/diagnostics12051248

**Published:** 2022-05-17

**Authors:** Man Soo Kim, Jae Jung Kim, Ki Ho Kang, Min Jun Kim, Yong In

**Affiliations:** Department of Orthopaedic Surgery, Seoul St. Mary’s Hospital, College of Medicine, The Catholic University of Korea, 222, Banpo-daero, Seocho-gu, Seoul 06591, Korea; kms3779@naver.com (M.S.K.); jaejung343@hanmail.net (J.J.K.); smilegiho@naver.com (K.H.K.); impbaba@naver.com (M.J.K.)

**Keywords:** central sensitization, pain, total knee arthroplasty, quantitative sensory test, central sensitization inventory, diagram, outcome

## Abstract

Central sensitization (CS) has been extensively researched as a cause of persistent pain after total knee arthroplasty (TKA). This systematic review study sought to investigate the diagnosis of CS in patients who underwent TKA for knee osteoarthritis (OA) and the effect of CS on clinical outcomes after TKA. Three comprehensive databases, including MEDLINE, EMBASE, and the Cochrane Library, were searched for studies that evaluated the outcomes of TKA in knee OA patients with CS. Data extraction, risk of bias assessment, and (where appropriate) meta-analysis were performed. The standardized mean difference (SMD) with a 95% confidence interval was used to assess the different scales of pain. A total of eight studies were selected, including two retrospective studies and five prospective observational studies. One study used additional randomized controlled trial data. Five studies were finally included in the meta-analysis. All studies had a minimum follow-up period of 3 months. The Central Sensitization Inventory (CSI), whole-body pain diagram, and quantitative sensory testing (QST) were used for measuring CS. The pooled analysis showed that patients with CS had more severe postoperative pain after TKA (SMD, 0.65; 95% CI, 0.40–0.90; *p* < 0.01) with moderate heterogeneity (I^2^ = 60%). In patients who underwent TKA with knee OA, CSI is most often used for the diagnosis of CS, and the QST and whole-body pain diagram are also used. CS is closely associated with more severe and persistent pain after TKA.

## 1. Introduction

Total knee arthroplasty (TKA) is a successful and well-established surgical treatment for end-stage knee osteoarthritis (OA) [1,2,3,4,5]. Most patients experience improvements in physical function and reduced pain after surgery. However, over the last decade, approximately 20% of patients have remained dissatisfied after TKA despite the substantial progress made in surgical equipment and techniques [6]. Various factors associated with patient dissatisfaction after TKA include the level of residual pain, functional outcome, and preoperative expectation [7]. However, persistent pain after TKA has the strongest association with patient dissatisfaction [8,9].

The International Association for the Study of Pain (IASP) defines chronic pain as pain lasting >3 months [10]. Therefore, pain that persists after this period can be regarded as chronic pain [11,12]. Approximately 10–34% of patients continue to complain of pain for 3 months to 5 years after TKA [13]. Persistent pain after TKA can be due to various surgical factors, including infection, malalignment, arthrofibrosis, instability, and loosening [14,15,16,17,18,19,20,21]. Therefore, determining the cause of persistent pain after TKA is important for those patients. However, despite the absence of abnormal findings in terms of surgical factors and having a physically well-functioning knee, some patients report persistent pain after surgery. Chronic pain conditions that increase pain sensitivity, such as central sensitization (CS), fibromyalgia, restless leg syndrome, and opioid-induced hyperalgesia [22], are also important factors.

Evidence, such as pathology or tissue damage, to explain the specific cause of persistent chronic pain is often lacking [23]. In general, the causes of persistent chronic pain tend to be classified based on the peripheral pain mechanism [24,25]. However, centralized pain is increasingly being considered a basis for pain patterns that cannot be explained by peripheral pain mechanisms [24,25]. Cases of chronic pain without clear nociceptive input and specific tissue damage can be regarded as CS [25,26]. A pain-control abnormality in the central nervous system (CNS) that responds more sensitively to pain and amplifies pain in patients with chronic pain is termed CS (i.e., central augmentation, central amplification, and centralized pain). CS can be defined as an amplification of neural signals within the CNS that causes pain hypersensitivity; [4] in other words, the brain and spinal cord “turn up the volume” in response to potentially unpleasant stimuli. Patients with these diseases commonly exhibit allodynia and hyperalgesia.

The purpose of this systematic review study was to investigate the diagnosis of CS in patients who underwent TKA for knee OA and discern the effect of CS on clinical outcomes after TKA. Moreover, if possible, we hoped to conduct a meta-analysis of the effects of CS on the clinical outcomes of TKA.

## 2. Methods

This systematic review was performed following the Preferred Reporting Items for Systematic Reviews and Meta-Analysis (PRISMA) guidelines [27].

### 2.1. Data and Literature Sources

Three comprehensive literature databases, including MEDLINE, EMBASE, and the Cochrane Library, were searched by two reviewers for studies published in English that evaluated the outcomes of TKA for knee OA with CS until February 2022. Search terms included (MeSH term “osteoarthritis” and key words “arthritis,” “osteoarthritis,” “osteoarthrosis,” “gonarthrosis,” and “gonoarthritis”) or (MeSH term “arthroplasty” and key words “replacement,” “joint replacement,” and “alloarthroplasty”), and (MeSH term “central nervous system sensitization” and key words “central sensitization,” “chronic pain,” “nociplastic pain,” and “widespread pain”). After the initial electronic search, manual searches of the reference lists and the bibliographies of identified articles, including relevant reviews and meta-analyses, were completed to identify relevant trials that the electronic search may have missed.

### 2.2. Study Selection

Two reviewers independently assessed the studies for inclusion according to the predefined selection criteria as follows. The study subjects were patients who underwent TKA for knee OA. The purpose of the study was to examine the impact of CS on postoperative outcomes following TKA. The study design included a minimum follow-up period of 3 months. Articles were original studies with full-text records available in English. Titles and abstracts were screened for relevance. In cases of uncertainty, the full article was evaluated to determine eligibility. Discrepancies were adjusted through discussion.

### 2.3. Data Extraction

Two reviewers independently extracted data from each study using a standardized data-extraction form. Disagreements were resolved by discussion, and those unresolved through discussion were reviewed by a third reviewer. The following variables were included: first author, publication year, number of participants, age, sex, study length, proportion of CS at baseline, how to measure CS, evaluation parameters, and most important results of the study. We attempted to contact the study authors to gather supplementary information when there were insufficient or missing data in the articles.

### 2.4. Risk of Bias Assessment

Two reviewers independently assessed the methodological quality of each study using the Newcastle–Ottawa scale [28] recommended by the Cochrane Non-randomized Studies Methods Working Group. For our research purposes, the star system on the Newcastle–Ottawa Scale, which assigns stars based on bias level, was modified to include only low, high, and unclear biases. Each study was reviewed according to three categories: the selection (four items); the comparability (two items); and the outcome (three items). Unresolved discrepancies between reviewers were resolved either by consensus or by consultation with a third investigator.

### 2.5. Statistical Analyses

The main outcome of this study was the postoperative pain level after TKA in either the CS or non-CS group. The outcomes (Western Ontario and McMaster Universities Osteoarthritis Index (WOMAC), pain visual analog scale (VAS) score) were calculated and presented as standardized mean differences (SMDs) with 95% confidence intervals (CIs) depending on the different scales used for the evaluation of pain. Five studies included postoperative pain assessments. Heterogeneity was determined using the I^2^ statistic, with values of 25%, 50%, and 75% indicative of low, moderate, and high heterogeneity, respectively. If I^2^ < 50%, pooled data were analyzed using a fixed-effects model; otherwise, a random-effects model was used. All statistical analyses were performed using RevMan version 5.3 (The Cochrane Collaboration, London, UK).

## 3. Results

### 3.1. Identification of Studies

Figure 1 shows the details of study identification, inclusion, and exclusion. An electronic search yielded 4188 studies in PubMed (MEDLINE), 9044 in EMBASE, and 1583 in the Cochrane Library. After removing 3796 duplicates, 11,019 studies remained; of these, 11,011 were excluded based on readings of the abstracts and full-text articles. After applying these criteria, eight studies were finally included in this systematic review, and five studies were finally included in the meta-analysis (Figure 1).

### 3.2. Study Characteristics

A total of eight studies were selected, including two retrospective studies [29,30] and five prospective observational studies [31,32,33,34,35]. One study used additional randomized controlled trial data [36]. All studies had a minimum follow-up period of 3 months (Table 1). The Central Sensitization Inventory (CSI), whole-body pain diagram, and quantitative sensory testing (QST) were used for measuring CS.

There were four studies that used the CSI [29,30,32,34] and two studies that used the whole-body pain diagram [31,33] and QST each [35,36]. In the case of the CSI, ≥40 points was defined as CS, and, in the case of the whole-body pain diagram, it was divided into three groups with 0, 1–2, and ≥3 pain sites, respectively. In the case of QST, criteria for clearly distinguishing CS were not presented, and one study compared the clinical results by distinguishing between low QST and high QST groups. All studies had a minimum follow-up period of 3 months. As a tool to measure postoperative clinical outcomes, WOMAC was used the most in six studies [29,30,31,33,35,36] (Table 2). All included studies showed a low risk of selection bias. Four studies provided detailed demographic data for the CS and non-CS groups [29,30,31,32], and four studies did not provide detailed demographic data for these groups [33,34,35,36]. Five of the eight studies were considered high-quality studies, with a score of ≥6 points on the Newcastle–Ottawa scale [29,30,31,32,35]. Only high-quality studies were finally included in the meta-analysis (Table 3).

**Table 1 diagnostics-12-01248-t001:** Characteristics of the included studies.

	Country	Design	Age (Years)	Number of Patients (Proportion of Female Patients)	Study Length	Study Population
Sasaki et al., 2022 [34]	Japan	Prospective observation study	71.5	40 (85.0%)	3 months	Improved group with CS Remained group with CS
Kim et al., 2021 [29]	Korea	Retrospective study	CS: 69.4 Non-CS:70	CS: 102 (86.3%) Non-CS: 320 (89.4%)	24 months	CS Non-CS
Lape et al., 2020 [33]	Korea	Prospective observation study	66.1 (8.5)	176 (63.6%)	12 months	Widespread pain groups (Painful body regions 0 vs. 1–2 vs. ≥3)
Koh et al., 2020 [30]	Korea	Retrospective study	70 (57–83)	Total 222 (91%) CS: 55 (91%) Non-CS:167 (90%)	24 months	CS Non-CS
Dave et al., 2017 [31]	USA	Prospective observation study	Pain site 0: 66.5 Pain sites 1–2: 65.6 Pain sites ≥ 3: 67.2	Pain site 0: 53 (64.1%) Pain sites 1–2: 121 (55.4%) Pain sites ≥ 3: 67 (67.2%)	12 months	Widespread pain groups (Painful body regions 0 vs. 1–2 vs. ≥3) Subgroup analysis compared the group with ≥3 painful body regions and the group with 0 painful body regions.
Waldy et al., 2015 [36]	England	Additional study using RCT data		239 (52.3%)	12 months	Patients who underwent TKA to measure widespread pain sensitivity through QST
Kim et al., 2015 [32]	Korea	Prospective observation study	CS: 69.2 Non-CS: 71.1	94 (100%)	3 months	CS Non-CS
Waldy et al., 2013 [35]	England	Prospective cohort (exploratory study)	68	51 (56.9%)	13 months	Knee OA patients with QST Healthy people without knee pain Comparison of lower QST group and higher QST group in patients with knee OA pain by subgroup analysis

CS, central sensitization; RCT, randomized controlled trial; TKA, total knee arthroplasty; QST, quantitative sensory testing; OA, osteoarthritis.

### 3.3. Diagnosis of CS

Two studies evaluated the CS in the patients who underwent TKA for knee OA using QST [35,36]. Electrical sensation, pressure pain threshold (PPT), and hot pain threshold (HPT) were measured in body parts locally and distantly from the affected knee joint to evaluate localized and widespread pain sensitization, respectively. In two studies, patients were investigated for widespread pain through whole-pain diagrams of painful areas throughout the body [31,33]. CS was evaluated by asking the patient to indicate areas with pain other than the knee joint.

In four studies, CS was diagnosed through CSI [29,30,32,34]. CSI is a 25-item questionnaire used to evaluate somatic and emotional symptoms, including headache, fatigue, sleep disorders, cognitive impairment, and psychological disorders, frequently observed in patients with CS, and pain sensitivity-related questions that could be experienced in daily life, including waking unrefreshed in the morning, stiff and achy muscles, anxiety attacks, grinding or clenching teeth, diarrhea and/or constipation, needing help in performing daily activities, sensitivity to bright lights, getting easily tired with physical activity, pain all over the body, headaches, feeling discomfort or burning during urination, poor sleep quality, difficulty in concentrating, skin problems, stress rendering physical symptoms worse, sadness or depression, low energy, muscle tension in neck and shoulders, pain in the jaw, dizziness and nausea caused by certain smells, frequent urination, uncomfortable and restless legs, poor memory, childhood trauma, and pelvic pain. Each item was scored on a 5-point Likert scale from 0–4 points (0 = never, 1 = rarely, 2 = sometimes, 3 = often, 4 = always) [29,30,32,34].

### 3.4. Clinical Manifestations Based on CS following TKA

Two studies compared clinical outcomes at 3 months after surgery [32,34]. One study compared whether CS was maintained or not at postoperative 3 months in CS patients and evaluated Knee Injury and Osteoarthritis Outcome Scale (KOOS) and EuroQoL-5-dimensions questionnaire (EQ-5D) results between these two groups. Patients with maintained CS before and after surgery reported inferior KOOS and EQ-5D results compared to patients with improved CS. Preoperative CS and postoperative EQ-5D scores were closely related [34]. The other study compared pain VAS scores, satisfaction, and pain relief at 3 months after surgery and reported that the CS group had more severe pain and less satisfaction and pain relief [32].

Three studies compared clinical outcomes at 1 year after surgery [31,33,35,36]. The study using QST was basically a study to assess the association between QST level and WOMAC pain at 1 year postoperatively [35,36]. However, subgroup analysis was performed in one study [35]. The results of 1 year of WOMAC pain were compared by dividing them into low QST and high QST groups. It was confirmed that the high QST group had more severe WOMAC pain at 1 year after surgery [35]. In the other two studies, WOMAC scores were compared by dividing into 3 groups according to pain site, not simply 2 groups, using pain diagrams [31,33]. In one study, there was no association between widespread pain and WOMAC pain [33], and, in another study, there was a close association between widespread pain before surgery and WOMAC pain, which was more severe at 1 year after surgery; the achievement rate of WOMAC minimum clinically important difference (MCID) was also significantly lower [31]. Moreover, in one study, when the group with 3–6 pain regions and the group with 0 pain regions were compared using subgroup analysis, the group with ≥3 pain regions showed inferior WOMAC scores and an MCID achievement rate [31]. A comparison of clinical results at 2 years after surgery was performed in two studies [29,30]. In the CS group, all subscores of WOMAC were inferior up to 2 years after surgery [29,30].

The pooled analysis showed that patients with CS have more severe postoperative pain after TKA (SMD, 0.65; 95% CI, 0.40–0.90; *p* < 0.01) with moderate heterogeneity (I^2^ = 60%) (Figure 2).

## 4. Discussion

This systematic review shows that preoperative CS has a close relationship with poor clinical features after surgery in patients who underwent TKA for knee OA. A meta-analysis also showed that the CS group experienced more severe and persistent pain after surgery than the non-CS group.

CS is a cause of severe and persistent pain after TKA, and many studies have been conducted [29,30,31,32,34,35,36,37,38,39,40]. CS is characterized by allodynia and hyperalgesia. Non-stimulatory pain means that even non-painful stimuli, such as light touches, cause pain. Hyperalgesia leads patients to feel painful stimuli to a greater intensity compared to normal people. One mechanism that causes excessive chronic pain is decreased activation of the descending anti-nociceptive pathway, which is most likely due to a deficiency in the pathway that responds to serotonin/norepinephrine. Central pain is characterized as dysfunctional in contrast to inflammatory, neurogenic, or structural pain, which are considered adaptive and potentially protective [4,5].

The QST is a popular method for evaluating the underlying mechanisms of CS. The QST determines sensory and pain thresholds for cold and warm temperatures as well as thresholds for vibration sensation through skin stimulation. These stimuli activate specific receptors that transmit messages to the CNS via peripheral nerve fibers. QST is a method to evaluate and quantify sensory function [41]. An increased sensitivity to sensory input from painless or healthy body parts based on the QST can be interpreted as a sign of CS. The QST can be performed at local (on or close to the site of the affected joint) or distant sites (remote site from the affected joint) using external stimulation to investigate the presence of CS [42]. Significant facilitations, such as positive temporal summation and decreased inhibition, in patients with knee OA yield findings suggestive of CS [43,44]. Widespread pain is also well known as a type of CS. In order to find out the widespread pain in various studies, the whole-pain diagram was used to approach the pain area in patients [25,31,33,45].

The CSI is a newly developed and validated self-reported inventory for evaluating patients with CS [46,47]. Unlike the QST, which assesses the response of the sensory system to sensory input, CSI primarily assesses patient-reported symptoms associated with CS. The CSI scale ranges from 0–100 points, with 0 points as the worst score and 100 points as the best score [46,47]. Neblett et al. [47] reported 40 points was a cutoff value for confirming CS based on the CSI score. CSI can be completed rapidly (<10 min) and easily without any equipment. In addition, the questionnaire includes non-painful and imaginable situations without ethical concerns. Therefore, these tests are useful for evaluating the severity of CS-related symptoms. CSI is a reliable and validated tool for quantifying the severity of various symptoms associated with CS [42,48].

In addition, magnetic resonance imaging (MRI) may additionally be used to diagnose CS. Functional neuroimaging using MRI provides important information on central pain mechanisms and shows whether structural and functional brain abnormalities exist in patients with CS [49,50,51]. Abnormal MRI findings include decreased cortical thickness; decreased brain volume; and increased levels of excitatory neurotransmitters, including glutamate [49,50,51].

This study confirmed that CS showed a close association with poor clinical outcomes after surgery in patients who underwent TKA for knee OA. In addition to the studies introduced in this study, there are various studies showing this fact [29,30,31,32,34,35,36]. Pain at rest and pain with movement were investigated in 69 patients who underwent TKA for knee OA, and their postoperative pain at 18 months after TKA was evaluated. Pain at rest was more closely associated with sensitization of nerve terminals in the dorsal horn and spinal cord neurons than pain with movement, consequently indicating CS. Patients with higher VAS scores for preoperative pain at rest showed significantly lower results in terms of postoperative pain relief [52]. Preoperative widespread pain sensitization measured using pressure algometry or electrical stimulation showed a close association with more severe pain after TKA [35,52].

There may be various reasons why CS is closely related to poor clinical outcomes after TKA. First, patients with CS tend to have higher preoperative expectations than patients without CS. Patients with CS had significantly higher expectations in terms of pain relief and psychological well-being than patients without CS [53]. High expectations can contribute to good postoperative outcomes [54]; however, expectations that are too high are closely associated with poor clinical outcomes after surgery [55]. Second, the clinical outcome after surgery is poor due to increased pain sensitivity in CS patients [56]. CS patients have a greater sensitivity to pain, and hyperalgesia and allodynia are characteristic findings [25]. The increased pain sensitivity in CS patients is also an important factor that can affect poor clinical outcomes after surgery [56]. Third, CS patients have greater MCID values than non-CS patients, and their postoperative clinical outcome is poorer and the rate of achieving an MCID is lower [29]. Due to these various causes, CS patients are more likely to experience persistent pain and inferior outcomes after TKA.

In CS patients who have persistent pain after TKA, treatment should be approached from various aspects. First, providing realistic expectations through appropriate education for patients before surgery is important. A close relationship exists between preoperative expectations of the patient and clinical outcomes after surgery. Therefore, when performing TKA in CS patients, discussing potential outcomes preoperatively can provide patients with realistic expectations of clinical outcomes after surgery. In addition, when TKA is performed for knee OA in a patient with CS, providing appropriate information regarding CS before surgery is necessary [57,58]. Second, appropriate use of medications can be helpful in CS. The selective serotonin norepinephrine reuptake inhibitor duloxetine appears to be effective, especially in CS patients with knee OA [59]. Koh et al. conducted a randomized controlled trial on the effects of duloxetine in CS patients who underwent TKA [38]. In the group administered duloxetine, significant pain reduction was observed from 2 weeks after surgery, and mood, mental health, enjoyment of life, relationships with others, and quality of sleep were significantly improved from 2 weeks after administration [38].

This study has several limitations. First, studies on the relationship between CS and post-TKA clinical features are still lacking in number. In this study, only studies written in English were included, and there was no survey of the gray literature. Therefore, a limited meta-analysis was performed. Second, the characteristics of the patients included in this study have sufficient heterogeneity. This can be a factor that can sufficiently influence the clinical outcomes after surgery. Third, studies included in this meta-analysis have important methodological limitations. Various evaluation scales were included in relation to postoperative pain, and various time periods were included. Lastly, since the number of patients included in the study is limited, additional studies are likely to be necessary in the future.

## 5. Conclusions

In patients who underwent TKA with knee OA, CSI is most often used for screening CS, and QST and pain diagrams are also used. CS is closely associated with more severe and persistent pain after TKA. Based on reviews, when performing TKA in CS patients, it is important to develop realistic patient expectations through appropriate education on general postoperative pain patterns in CS.

## Figures and Tables

**Figure 1 diagnostics-12-01248-f001:**
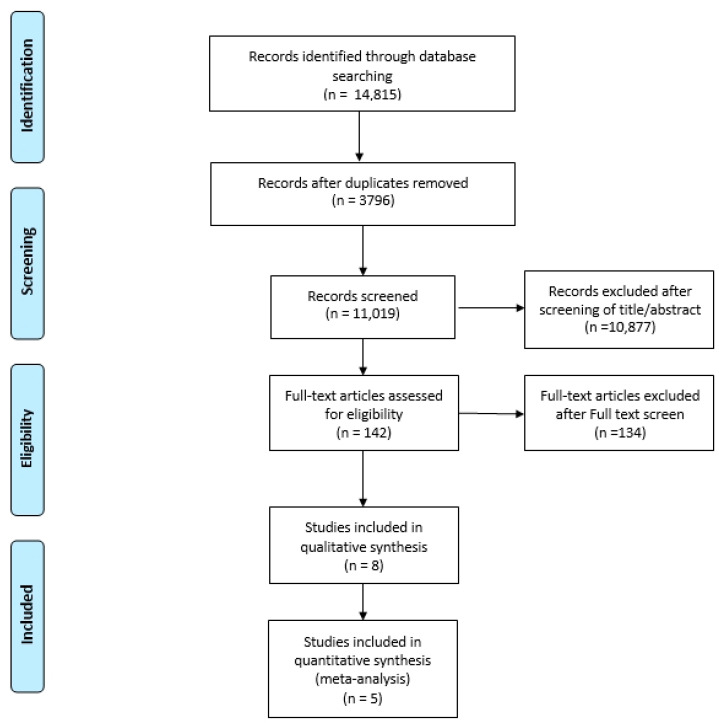
PRISMA flow diagram for the systematic review.

**Figure 2 diagnostics-12-01248-f002:**
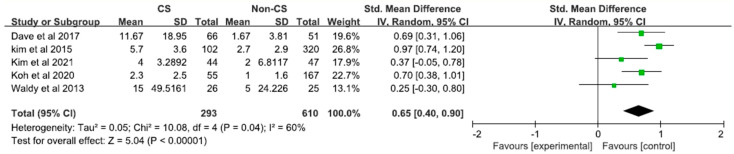
Forest plot of the relationship between CS and postoperative pain after TKA [29,30,31,32,35].

**Table 2 diagnostics-12-01248-t002:** The diagnosis of CS and the relationship between CS and postoperative outcomes after TKA.

Study	Proportion of CS at Baseline	Measure of CS	Postoperative Outcome Measure	Important Results and Comments
Sasaki et al., 2022 [34]	19(47.5%)	CSI	KOOS EQ-5D	Preoperative CS was negatively associated with EQ-5D score after TKA (β = −0.44, *p* = 0.017) Patients who maintained CS before and after surgery had inferior KOOS/EQ-5D results compared to those who improved (all *p* < 0.05)
Kim et al., 2021 [29]	102 (24.2%)	CSI	WOMAC	The CS group showed significantly inferior preoperative and postoperative WOMAC pain, function, and total scores compared to the non-CS group (all *p* < 0.05) Preoperative WOMAC total score: CS 61.0 vs. non-CS 57.1 (*p* < 0.05) Postoperative WOMAC total score: CS 25.8 vs. non-CS 17.4 (*p* < 0.05) Preoperative WOMAC total score: CS 13.6 vs. non-CS 11.9 (*p* < 0.05) Postoperative WOMAC total score: CS 5.7 vs. non-CS 2.7 (*p* < 0.05)
Lape et al., 2020 [33]		Whole-body pain diagram (19 sites labeled on the diagram)	WOMAC	There was no significant association between changes in the widespread pain groups and changes in WOMAC pain scores (*p* > 0.05).
Koh et al., 2020 [30]	55 (24.8%)	CSI	Pain VAS WOMAC KSS Satisfaction (new KSS)	The CS group showed worse quality of life, functional disability, and dissatisfaction than the non-CS group after TKA (all *p* < 0.05). Postoperative pain VAS score: CS 2.3 vs. non-CS 1.0 (*p* < 0.05) Postoperative WOMAC total score: CS 25.2 vs. non-CS 15.4 (*p* < 0.05) Postoperative KSS total score: CS 165.3 vs. non-CS 177.6 (*p* < 0.05)
Dave et al., 2017 [31]		Whole-body pain diagram (19 sites labeled on the diagram)	WOMAC MCID	Preoperative widespread pain was associated with greater pain at 12 months and failure to reach the MCID (All *p* < 0.05) Patients with preoperative pain in 3–6 body regions showed higher WOMAC scores at follow-up compared to patients with no painful body regions (median, 10 vs. 0) and were also less likely to achieve MCID (77% vs. 98%) (all *p* < 0.05)
Waldy et al., 2015 [36]		QST (PPT)	WOMAC	There was no definite association between preoperative PPTs and pain severity at 12 months after TKA in any of the linear regression models (All *p* < 0.05)
Kim et al., 2015 [32]	44 (46.8%)	CSI	VAS Satisfaction (pain relief, functional improvement)	Postoperative pain VAS score: CS 4 vs. non-CS 2 (*p* < 0.05) CS patients reported poor satisfaction regarding pain relief compared to non-CS patients (*p* < 0.05)
Waldy et al., 2013 [35]		QST (PPT and HPT)	WOMAC	When patients were divided into low and high preoperative forearm PPT groups, patients in the low PPT group showed worse 1-year WOMAC pain scores compared to patients in the high PPT group (85 vs. 95, *p* < 0.05)

CS, central sensitization; CSI, Central Sensitization Inventory; KOOS, Knee Injury and Osteoarthritis Outcome Scale; EQ-5D, EuroQoL-5-dimensions questionnaire; TKA, total knee arthroplasty; WOMAC, Western Ontario and McMaster Universities Arthritis Index questionnaire; VAS, visual analog scale; KSS, Knee Society Score; MCID, minimum clinically important difference; QST, quantitative sensory testing; PPT, pressure pain threshold; HPT, hot pain threshold.

**Table 3 diagnostics-12-01248-t003:** Risk-of-bias assessment performed using the Newcastle–Ottawa scale.

Quality Assessment of the Studies by the Newcastle–Ottawa Scale										
	Selection				Comparability		Outcome			
Study	Representative of the Cases	Selection of Control	Ascertainment of Exposure	Outcome of Interest Not Present at the Start of the Study	Comparability of Cohorts	Control for Any Additional Factor	Assessment of Outcomes	Sufficient Follow-Up	Adequacy of Follow-Up	Total 9/9
Sasaki et al. [34]	*	0	*	*	0	0	*	*	0	5
Kim et al. [29].	*	*	*	*	*	0	*	*	0	7
Lape et al. [33]	*	0	*	*	0	0	0	*	*	5
Koh et al. [30]	*	*	*	*	*	0	*	*	0	7
Dave et al. [31]	*	*	*	*	*	*	*	*	*	9
Waldy et al. [36]	*	0	*	*	0	0	*	*	0	5
Kim et al. [32]	*	*	*	*	*	*	*	*	*	9
Waldy et al. [35]	*	0	*	*	*	0	*	*	*	7

* = low risk; 0 = high risk.

## Data Availability

The data presented in this study are available in the main article.

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
