# Peer review of "Diagnosis of Central Sensitization and Its Effects on Postoperative Outcomes following Total Knee Arthroplasty: A Systematic Review and Meta-Analysis"

_diagnostics, 2022, doi:10.3390/diagnostics12051248_

Round 1

Reviewer 1 Report

This is an interesting systematic review and meta-analysis. The authors aimed to investigate the diagnosis of central sensitization in patients who underwent total knee arthroplasty for knee osteoarthritis and its effect on clinical outcomes.

The study is well designed according to PRISMA guidelines, and the methods and results are well presented. The discussion is interesting and well written.

The authors changed the manuscript according to the reviewers' suggestions. I think that this review can be accepted in the present form. 

Author Response

This is an interesting systematic review and meta-analysis. The authors aimed to investigate the diagnosis of central sensitization in patients who underwent total knee arthroplasty for knee osteoarthritis and its effect on clinical outcomes.

â–¶We thank the reviewer for his/her valuable time and we agree with this succinct summary of our study.

The study is well designed according to PRISMA guidelines, and the methods and results are well presented. The discussion is interesting and well written.

â–¶Thank you for your comments.

The authors changed the manuscript according to the reviewers' suggestions. I think that this review can be accepted in the present form. 

â–¶Thank you for your comments.

Reviewer 2 Report

Interesting article, focused on an important and current topic. The manuscript is well written, in a good english, and the study is clearly explained. I think the paper is suitable for publication.

Author Response

Interesting article, focused on an important and current topic. The manuscript is well written, in a good english, and the study is clearly explained. I think the paper is suitable for publication.

â–¶We thank the reviewer for his/her valuable time and we agree with this succinct summary of our study.

Reviewer 3 Report

I liked reading the manuscript and it is an important field to make a review.

However, is this a revision process?

Otherwise, can the authors provide a clean manuscript for the review process, it would make reading easier and looks more professional. 

Author Response

I liked reading the manuscript and it is an important field to make a review. However, is this a revision process? Otherwise, can the authors provide a clean manuscript for the review process, it would make reading easier and looks more professional. 

â–¶Thank you for your comments. This manuscript was provided in revision format, but almost all contents have been changed. The previous manuscript was written in narrative review format. But, by the requests of reviewers, it was rewritten in systematic review format according to the PRISMA guideline.  Thanks again for your advice.

Round 2

Reviewer 3 Report

The authors made the requested changes

Accept.